# Opioid Free Anesthesia in Thoracic Surgery: A Systematic Review and Meta Analysis

**DOI:** 10.3390/jcm11236955

**Published:** 2022-11-25

**Authors:** Filippo D’Amico, Gaia Barucco, Margherita Licheri, Gabriele Valsecchi, Luisa Zaraca, Marta Mucchetti, Alberto Zangrillo, Fabrizio Monaco

**Affiliations:** 1Department of Anesthesia and Intensive Care, IRCCS San Raffaele Scientific Institute, 20132 Milan, Italy; 2Faculty of Medicine, Vita-Salute San Raffaele University, 20132 Milan, Italy

**Keywords:** opioid-free anesthesia, thoracic surgery, post-operative complication, opioid consumption, opioid-free analgesia, Enhanced Recovery After Surgery (ERAS)

## Abstract

Introduction: Recent studies showed that balanced opioid-free anesthesia is feasible and desirable in several surgical settings. However, in thoracic surgery, scientific evidence is still lacking. Thus, we conducted the first systematic review and meta-analysis of opioid-free anesthesia in this field. Methods: The primary outcome was the occurrence of any complication. Secondary outcomes were the length of hospital stay, recovery room length of stay, postoperative pain at 24 and 48 h, and morphine equivalent consumption at 48 h. Results: Out of 375 potentially relevant articles, 6 studies (1 randomized controlled trial and 5 observational cohort studies) counting a total of 904 patients were included. Opioid-free anesthesia compared to opioid-based anesthesia, was associated with a lower rate of any complication (74 of 175 [42%] vs. 200 of 294 [68%]; RR = 0.76; 95% CI, 0.65–0.89; *p* < 0.001; I^2^ = 0%), lower 48 h morphine equivalent consumption (MD −14.5 [−29.17/−0.22]; *p* = 0.05; I^2^ = 95%) and lower pain at 48 h (MD −1.95 [−3.6/0.3]; *p* = 0.02, I = 98%). Conclusions: Opioid-free anesthesia in thoracic surgery is associated with lower postoperative complications, and less opioid demand with better postoperative analgesia at 48 h compared to opioid-based anesthesia.

## 1. Introduction

Analgesia is a major determinant of balanced anesthesia and it is usually achieved by administering opioid agents, which are well tolerated and maintain hemodynamic stability in the perioperative period [1]. However, perioperative opioid administration is not risk-free. Opioids are associated with life-threatening side effects such as respiratory depression, postoperative nausea and vomiting, opioid-induced hyperalgesia, constipation, urinary retention, immunomodulation and neurotoxicity [2]. In addition, opioid prescription after surgery seems to trigger the development of opioid addiction, thus contributing to the widespread opioid misuse observed worldwide [3]. Several studies found a correlation between postoperative opioid administration, the development of chronic pain and opioid addiction [4].

Therefore, opioid administration should be reduced or avoided as much as possible. There is growing evidence showing that opioid-free anesthesia (OFA), including loco-regional anesthesia and non-opioid drugs, is feasible in several surgical settings. However, thoracic surgery is a much more challenging field associated with a higher rate of pain and pulmonary complications compared to other surgeries. Adequate analgesia after thoracic surgery is essential for a successful outcome. Indeed, both early mobilization and the ability to cough after surgery are key elements to decrease the risk of postoperative pneumonia, atelectasis and respiratory failure.

Experience gained in other surgeries cannot be directly translated into thoracic surgery, which needs own considerations. Previous studies on OFA in thoracic surgery are heterogeneous with small sample sizes, with possible risk of Type II statistical error. In conclusion, whether OFA in thoracic surgery may be effective or not is still uncertain [5,6].

We performed a systematic review and meta-analysis aimed at summarizing current evidence on opioid-based anesthesia (OBA) versus OFA in thoracic surgery. Our primary endpoint was the rate of postoperative complications. We, then, considered length of hospital stay (LOS), recovery room length of stay, postoperative pain at 24 and 48 h, and morphine equivalent consumption at 48 h as secondary endpoints.

## 2. Materials and Methods

We conducted this systematic review and meta-analysis according to the Preferred Reporting Items for Systematic Reviews and Meta-Analyses guidelines [7]. Following the PICO (Population, Intervention or exposure, Comparison, Outcome) framework [8] the review question was: In patients undergoing thoracic surgery (P), does opioid-free anesthesia (E), compared to opioid anesthesia (C), reduce complication (O)? This review was registered in PROSPERO (CRD 42022344504).

### 2.1. Search Strategy and Selection Criteria

We performed a systematic search on PubMed and EMBASE up to 1 July 2022. Keywords and other free terms were used with Boolean operators (OR, AND) to combine searches:

((opiod-free [tiab]) or (opioid-sparing [tiab]) or (regional anesthesia [mesh]) or (block [tiab]) or (local anesthetic [tiab]) or (ketamine [tiab])or (dexmedetomidine [tiab]) or (bupivacaine [tiab]) or (ropivacaine [tiab]) or (lidocaine [tiab])) AND (Thoracic surgery [tiab] or (lobectomy [tiab]) or (bilobectomy [tiab]) or (segmentomy [tiab]) or (pneumectomy [tiab])) and ((randomized controlled trial [Publication Type] OR (randomized [Title/Abstract] AND controlled [Title/Abstract] AND trial [Title/Abstract]) or (retrospective [tiab]) or (case-report [tiab]) or (case-series [tiab]) or (observational [tiab])))

Further studies were selected by manual research of the references identified from the original studies. We employed backward snowballing (scanning through references of retrieved articles and pertinent reviews) and when necessary we contacted international experts to obtain data missing from the original paper. No language restrictions were enforced.

We included studies comparing OFA with OBA in patients undergoing thoracic surgery. Randomized controlled trials and observational cohort studies published in peer-reviewed journals were considered eligible. Studies reporting patients of any setting, systematic and narrative reviews, and editorials were excluded. After the removal of duplicates, two authors independently assessed compliance with selection criteria at the title/abstract level using a standardized form with disagreements resolved by consensus or by involving a third investigator if required. The final selection of included articles was based on complete manuscripts with disagreements resolved by consensus. Study characteristics (first author, year of publication, country), sample size, complication, postoperative opioid consumption, post-operative pain, LOS were extracted and reported.

### 2.2. Outcomes

The primary outcome was the occurrence of any complication following the Devin’s classification of surgical complications [9]. We considered overall postoperative complication grade II or more, definite as any complication requiring pharmacological treatment non commonly used in the postoperative setting, surgical, radiological, endoscopic intervention, organ dysfunction, or life-treating complication. Secondary endpoints were LOS, postoperative pain at 24 and 48 h, and morphine equivalent consumption at 48 h. To allow the comparison, pain scores reported as visual, verbal, or numeric rating scales were converted to a standardized 0–10 analog scale for quantitative evaluation. When scores were not presented in a 0–10-point-scale format, they were converted (mean and SD) using the appropriate ratio. In particular, when reported on a scale of 0–3, we consider 2 as the medium score (5), 0 as a minimum (0), and 3 as a maximum (10).

### 2.3. Statistical Analysis

Computations were performed with Review manager version 5.4.1. This meta-analysis was performed in compliance with PRISMA (Preferred Reporting Items for Systematic reviews and Meta-Analyses) [10]. We calculated pooled risk ratio (RR) for the primary and secondary outcomes and 95% confidence intervals (CI) using the Mantel-Haenszel method for dichotomous outcomes [11]. For the continuous outcome, the mean difference (MD) and 95% CIs outcomes with the inverse variance (I–V) method were used. If only a median and interquartile range (IQR) were available, Wan’s method was used to estimate the mean and standard deviation (SD) [12]. The statistical heterogeneity hypothesis was evaluated with statistical significance set at the two-tailed 0.05 levels, whereas the extent of statistical consistency was quantified with Higgins and Thompson’s I^2^. I^2^ values around 25, 50, and 75% were considered respectively low, moderate, and severe statistical inconsistency (I^2^ > 50% was used as a threshold indicating significant heterogeneity for individual studies) [13]. Pooled data were analyzed using the inverse variance method with a fixed-effect model in case of low-moderate (I^2^ < 50%) statistical inconsistency, or with a random-effect model when the I^2^ was above 50% [8]. A *p*-value less than 0.05 was considered statistically significant. The risk of bias was appreciated by the tool Risk Of Bias In Nonrandomized Studies—of Interventions (ROBINS-I) [14]. The funnel plot and the Egger’s test asymmetry were not performed due to both a reduced number of studies available (<10) and the small sample size [15]. A sensitivity analysis was performed to take into account the effect of different type of surgery on the outcomes and to address the heterogeneity.

## 3. Results

### 3.1. Study Characteristics

The research strategy of electronic databases detected 375 potentially relevant articles. Six studies with a total of 904 patients met the inclusion criteria and were included (Figure 1). All studies were conducted between May 2016 and July 2019. Three studies were conducted in Europe [16,17,18], one in Australia [19], one in the USA [20], and one in China [21] (Table 1). Only one study was randomized controlled [21], while the others were observational retrospective studies. All trials were monocentric and included patients undergoing minimally invasive surgery (robotic or video-assisted thoracic surgery) except for Bello et al. [16] and Devine et al. [19] in which the entire population and 33/187 patients respectively, underwent open surgery. All studies used loco-regional analgesia (thoracic epidural analgesia, paravertebral block, thoracic wall blocks). Most studies used remifentanil (*n* = 33%) [16,17], one used sufentanil [18], while the remaining studies considered more than one opioid as comparator (Table 1). In the OFA group opioids were never used even during general anesthesia induction.

### 3.2. Risk of Bias

Application of the ROBINS-I tool suggested that the majority of trials had a low risk of bias (Figure 2). Overall, all trials were classified as being at low risk of bias. The randomization was adequately described in the only RCT included in the meta-analysis.

### 3.3. Complications

OFA showed a significant reduction in the postoperative complication rate compared to OBA (Figure 3. OFA vs. OBA 74/175 [42%] vs. 200/294 [68%]; RR = 0.76; 95% CI: 0.65–0.89; *p* < 0.001; I^2^ = 0%) [16,17,20]. Three studies reported data on the LOS that was shorter in OFA than in OBA express in days (Figure 4. MD = −0.95 [−1.23/−0.66]; *p* < 0.001; I^2^ = 0%) [18,19,20]. Recovery room length of stay was similar between the two groups express in hours (MD = −1.89 [−11.5/7.74]; *p* = 0.7; I = 50%) [19,21].

### 3.4. Morphine Equivalent Consumption

Post-operative oral morphine equivalent (OME) express in milligrams consumption was lower in OFA than in OBA. Three studies reported data on 48-h morphine equivalent consumption, with a decreased consumption in the OFA group than in the OBA (Figure 5. MD −14.5 [−29.17/−0.22]; *p* = 0.05; I^2^ = 95%) [16,17,18]. One study reported data on postoperative 24-h PCA morphine consumption that was similar between the two groups (16.2 ± 18.1 mg vs. 21.1 ± 18.8 mg; *p* = 0.16) [19]. An et al. failed to record data on opioid consumption after surgery [21]. Bello et al. found that fewer patients in the OFA group required morphine in recovery room (4% vs. 42%; *p* < 0.001) and morphine patient-controlled analgesia in ICU (12% vs. 22%; *p* = 0.01) than in the OBA group [16].

### 3.5. Pain Score

Three studies assessed analgesia at 24 h and two studies at 48 h after surgery. We found no statistically significant difference between the groups at 24 h (Figure 6; MD −1.69 [−3.82, 0.43]; *p* = 0.12; I^2^ = 100%) [16,17,19], whereas a significant reduction in the OFA group at 48 h (Figure 7; MD −1.95 [−3.6/0.3]; *p* = 0.02; I^2^ = 98%) [16,17]. One study showed a statistically significant reduction in pain scores during the hospital stay [20]. An et al. found that the differences were statistically non-significant between the two groups (*p* = 0.5) in the intraoperative analgesia assessed by the pain threshold index calculated from changes in electroencephalographic signals [21].

### 3.6. Sensitivity Analysis

The studies of Bello [16] and Devine [19] were the only two trials with an open surgical technique, while the others were minimally invasive. Since the surgical technique might, at least partially, affect the outcomes a second analysis was performed not considering, these trials, one at time. When the study of Bello [16] was excluded, the sensitivity analysis showed no substantive difference from the primary analysis in terms of complications and morphine consumption at 48 h (Appendix A). Unlike the primary analysis, the sensitivity analysis showed only a trend in favour of a shorter length of stay in the OFA compared with OBA group when the study of Devine [19] was excluded (Appendix A). Regarding the pain score at 24 h, the sensitivity analysis conducted without the Salim trial [17], which was the only one with minimally invasive technique among the studies reporting this outcome, led to the same results of the primary analysis (Appendix A). Finally, when the sensitivity analysis was conducted to explore the heterogeneity, we found that the source of heterogeneity for the morphine equivalent consumption at 48 h was the study of Selim [17] (Appendix A Appendix A). On the contrary, the sensitivity analysis led to the same result of the primary analysis for the other outcomes with high test of heterogeneity.

## 4. Discussion

The main findings of the present systematic review and meta-analysis are that OFA compared to OBA is associated with a lower rate of postoperative complications and a better analgesia with less opioid consumption at 48 h after surgery. To our knowledge, this systematic review and meta-analysis provides the strongest evidence on OFA in thoracic surgery available so far. Indeed, previous published reviews on OFA involved different surgical patients and show heterogeneous results. Current evidence remains therefore controversial. Frauenknecht et al. [22] showed that OFA and OBA are equivalent strategies in terms of postoperative pain control at 24 h after surgery. However, a lower opioid request was registered in the opioid-inclusive strategy group [22]. On the contrary, Salome et al. found that OFA is associated with a reduction in the pain score at 2 h [−0.75 (−1.18, −0.32)], followed by reduction in morphine administration at 2 and 24 h after surgery [MD: −1.61 (−2.69, −0.53) and −1.73 (−2.82, −0.65) (*p* < 0.05)] [23]. Similarly, also Fletcher described a decreased opioid consumption and postoperative pain at 24 h in the OFA group [24]. Despite statistically significant, these results are not clinically relevant. Indeed, the mean difference between the groups was 3 cm on a 100 cm visual analog scale (95% CI: 0.4–5.6, *p* = 0.02) and 0.7 mg for morphine consumption (95% CI: 0.37–299 1.02; *p* < 0.001) [24]. Notably, no patient undergoing thoracic surgery was enrolled in any mentioned metanalyses. This aspect should be highlighted, as particularly in thoracic surgery suboptimal analgesia is associated with increased morbidity and mortality [25]. Indeed, thoracic surgery patients often have a limited respiratory reserve, which combined with the opioid induced respiratory depression, renders this population at high risk of hypoventilation and iatrogenic injury in the perioperative period. Regarding the most common causes of acute pain within 24 h after surgery, Lee et al. reported that almost one-third of acute pain claims registered in the USA between 1990 and 2009, were associated with respiratory depression [26]. Therefore, we suggest clinicians to support and improve opioid-free regimens to minimize the respiratory depression risk. Once again, thoracic surgery could be the ideal setting for OFA, as it mostly involves mini-invasive surgical techniques (VATS or RATS) and efficient loco-regional anesthesia procedures. Nevertheless, there are only few studies on OFA in thoracic surgery and there is need for further evidence before suggesting a widening use in clinical practice. The present systematic review and meta-analysis aims to partially fill this gap of knowledge. We found that patients undergoing thoracic surgery met lower serious surgical perioperative complications when receiving OFA instead of OBA. Complications were never considered in previous meta-analyses on this topic except for nausea and vomiting. However, particularly thoracic surgery patients may benefit from OFA to improve short-term outcome. Our hypothesis is that both opioid restriction and adequate analgesia, enhance patient cooperation, ventilation and early mobilization. Particularly, mobilization within 24 h after surgery, is the most predictive parameter for a decreased 30 day morbidity [27]. Interestingly, the Thoracic Enhanced Recovery After Surgery (ERAS) Program strongly recommends a reduction in opioid use to improve outcome [28], even if the scientific evidence is low. Thus, the results of the current systematic review and meta-analysis may be included in future guidelines providing the first level of evidence on this hot topic.

Indeed, the recovery room LOS was not affected by the pharmacological protocol. This result is not surprising as the time spent in recovery room is multifactorial and the surgical or organizational factors may play a pivotal role respect with the analgesic regimen. It’s authors’ opinion that it would be a more reliable marker for further investigations to consider the “readiness from discharge” rather than the recovery room length of stay.

Whether OFA also affects the long-term outcome in a patient undergoing thoracic surgery for cancer lung resection is still under investigation. There is a growing body of evidence on the pleiotropic effects of opioids on cancer and survival after cancer surgery [29]. Opioids are believed to depress the immune system, stimulate angiogenesis and induce tumor growth by facilitating metastatic regrowth [30]. Therefore, OFA may have a beneficial impact on the long-term prognosis of patients undergoing thoracic surgery by reducing the exposition to opioids. However, this field deserves further investigations and it is beyond the scope of the current review.

The present study has strengths and limitations. Firstly, we studied a specific population, which only involved patients undergoing thoracic surgery, while previous studies involved various surgery patients Secondly, postoperative complications represent an original clinical outcome among meta-analyses on OFA, since previous studies analyzed as primary outcome opioid consumption and postoperative pain control. Moreover, we used a standard scoring system of complications, thus reducing the heterogeneity of definitions commonly observed among the studies allowing a more objective comparison of the outcome. The use of this standardized scoring system should be encouraged for future trials.

A limitation of the study was the high heterogeneity (I^2^ > 50%) of morphine consumption and pain score at 48 h. In order to investigate the source of heterogeneity we performed a sensitivity analysis which showed for the former that the heterogeneity depended on the study of Selim et al. [17] and for the latter from other unclear sources. This sounds as a call to run large randomized controlled trials on OFA in thoracic surgery.

Another limitation is the quality of the studies included in the meta-analysis. There was only one monocentric randomized controlled trial whereas the other studies were retrospective single center observational studies. In addition, inclusion of studies with less than 50 patients each, could lead to concern in the estimation of the treatment effect. The low number of studies also does not allow the use of tests such as funnel plots, which would permit a more accurate analysis of bias. We cannot exclude that the number of patients considered did not reach the optimal sample size to draw strong information (Type-II error). Our data, however, might fuel new methodologically robust prospective trials. Another possible source of bias could be the heterogeneous manner of reporting pain score among studies. However, converting the value to a reference scale has been widely validated previously [22,31,32,33].

In order to avoid the possible bias due to the influence of surgical techniques on outcomes, we perform a sensitivity analysis excluding the Bello [16] and Devine trials [19] once at time. Indeed, the result of the sensitivity analysis showed no substantive differences when compared with the primary analysis in terms of pain score at 24 h and morphine equivalent consumption at 48 h. Differently from the primary analysis, the sensitivity analysis conducted using only studies with minimally invasive surgical technique and excluding the Davine, investigation [19] in which less than 20% of patients received a thoracotomy, showed a trend to a shorter LOS in favor of OFA. It is reasonable that, as the minimally invasive settings are associated with lower inpatient stay than an open approach [34], further decrease of LOS is hard to achieve by pharmacological protocols. In practice, the surgical technique affects LOS much more than the pharmacological strategies.

Although the surgical approach (minimally invasive or open) may be a source of bias, the result of the sensitivity analysis showed no substantive difference from the morphine consumption at 24 h and pain score at 48 h when the study of Bello et.al [16] and Devine et al. [19] were excluded. Indeed, there was a discrepancy between the primary and the sensitive analysis only for LOS. In fact, the exclusion of the Devine study [19], which includes 33 with open thoracotomy on 187 overall patients, may have mitigated the effect of OFA on LOS. However, the limited number of studies currently available in the literature, makes hard to draw univocal conclusions between OFA and kind of surgical techniques.

Although we performed a systematic and extensive search on OFA in thoracic surgery in the largest databases available and by backward snowballing, we found only one RCT and 5 non-RCTs. Even if the combination of these types of studies in the same metanalysis may be a source of bias the results of the only RTC were not included in the metanalysis, as it reported outcome not direct comparable with the others.

Nevertheless, the present systematic review and metanalysis, based on small investigations conducted by academic independent researchers, is the first one available so far on this “actual topic” in thoracic surgery and it may provide a first level of evidence.

## 5. Conclusions

In conclusion, OFA might decrease postoperative complications, and opioid consumption at 48 h allowing a better postoperative analgesia compared to OBA. It is unlike that OFA may further decrease the LOS in patients undergoing minimally invasive surgery.

We believe that OFA it may be a promising strategy although multicenter randomized controlled trials studying up to date OFA in thoracic surgery are needed.

## Figures and Tables

**Figure 1 jcm-11-06955-f001:**
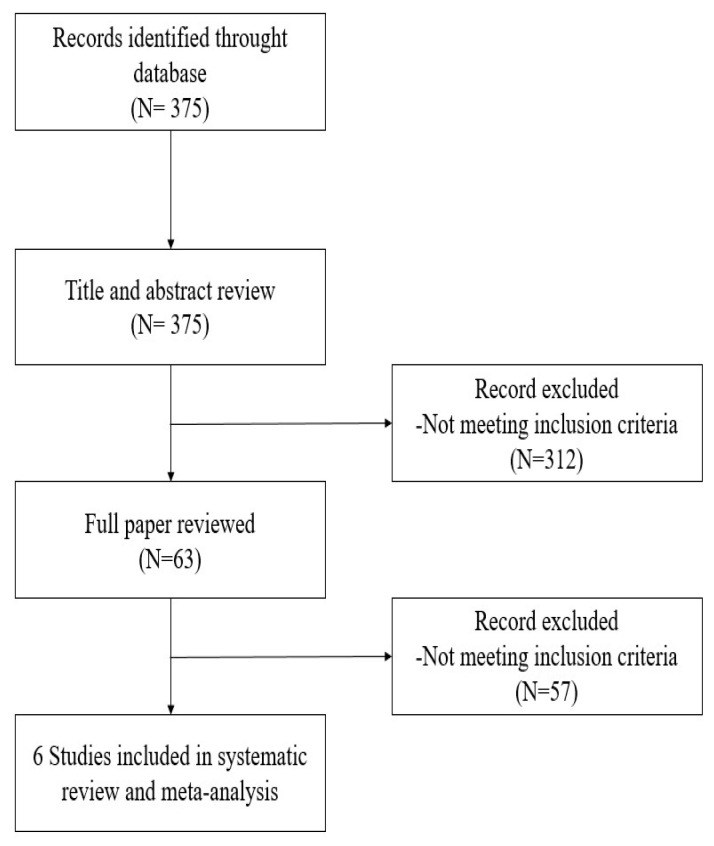
PRISMA flow diagram showing literature search results. Six trials were included in the analysis.

**Figure 2 jcm-11-06955-f002:**
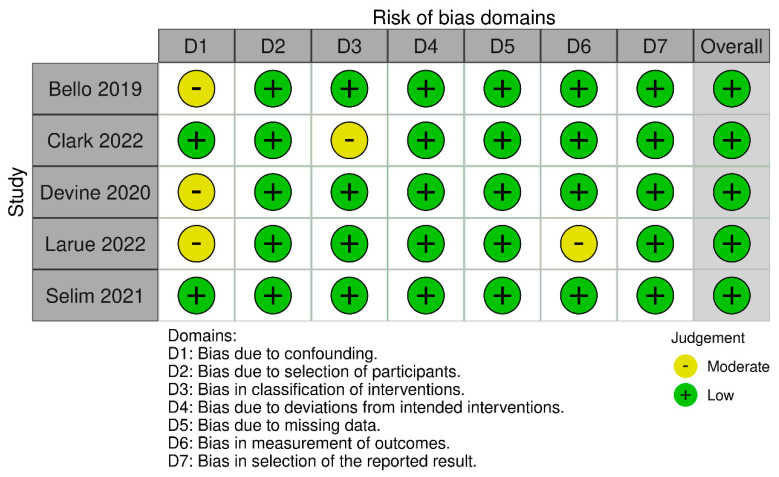
ROBINS−I evaluation of included studies. ROBINS−I risk of bias in non−randomised studies of interventions [16,17,18,19,20].

**Figure 3 jcm-11-06955-f003:**
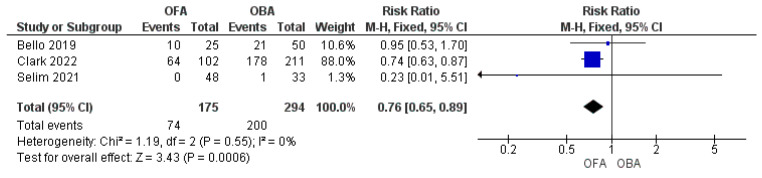
Forest plot of risk ratio to develop complications comparing opioid-free anesthesia and opioid-based anesthesia [16,17,20].

**Figure 4 jcm-11-06955-f004:**
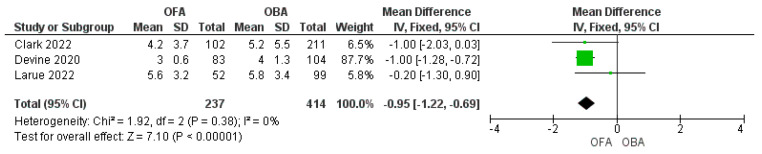
Forest plot of mean difference of length of stay comparing opioid−free anesthesia and opioid−based anesthesia [18,19,20].

**Figure 5 jcm-11-06955-f005:**
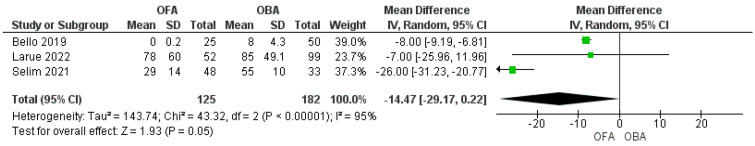
Forest plot of mean difference of morphine equivalent consumption at 48 h comparing opioid−free anesthesia and opioid−based anesthesia [16,17,18].

**Figure 6 jcm-11-06955-f006:**
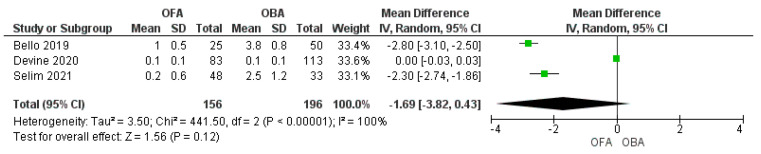
Forest plot of mean difference of pain at 24 h comparing opioid−free anesthesia and opioid−based anesthesia [16,17,19].

**Figure 7 jcm-11-06955-f007:**
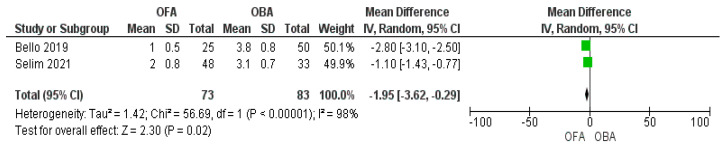
Forest plot of mean difference of pain at 48 h comparing opioid−free anesthesia and opioid−based anesthesia [16,17].

**Table 1 jcm-11-06955-t001:** Trial characteristics.

Study	Nation	Surgical Procedure	Intraoperative Analgesic Regimen		Sample Size OFA vs. OBA	Primary Outcome	Secondary Outcome
			OBA	OFA			
**Bello, 2019 [16]**	France	Open thoracotomy	TEA Remifentanil Ketamine	TEA Ketamine	25 vs. 50	Ropivacaine cumulative administration during the first 48 h post-operatively	- NRS at 6, 24 and 48 h - Number of patients with NRS > 4 - Morphine consumption in PACU - Total opioid at 48 h - PONV - Intraoperative hypertension (SBP > 150 mmHg) - Vasopressors - Postoperative complication
**Clark, 2022 [20]**	USA	Minimally invasive lobectomy	NR	INB or SAPB Acetominophen Gabapentin	102 vs. 211	In-hospital opioid consumption	- NRS - Discharge from the Hospital with opioid - Several surgical and medical post-operative outcomes
**Devine, 2020 [19]**	Australia	VATS and open thoracotomy	SAPB or PVB Fentanyl Remifentanil and/or morphine paracetamol and parecoxib	PVB Lidocaine Magnesium Paracetamol Parecoxib	83 vs. 104	Postoperative pain scores at 0, 1 and 24 h	- PCA morphine consumption - Recovery LOS - Hospital LOS - In hospital mortality - Thirty-day mortality
**An, 2021 [21]**	China	VATS	PVB Sufentanil Remifentanil	PVB Dexmedetomidine Ketorolac	49 vs. 48	Intraoperative PTI	- Wavelet index - Blood glucose - MAP, HR, lactic acid
**Larue, 2022 [18]**	France	VATS	SAPB/ESPB/PVB Sufentanil Ketamine Magnesium Desametasone	SAPB/ESPB/PVB Dexmetedonimine Ketamine Magnesium Desametasone	52 vs. 99	Opioid consumption at 48 h	Postoperative NRS
**Selim, 2021 [17]**	France	VATS or RATS	PVB/SAPB Remifentanil Ketamine Nefopam Ketoprofen	PVB/SAPB Dexmedetomidine Lidocaine Ketamine Nefopam Ketoprofen Paracetamol	48 vs. 33	Postoperative consumption of morphine within the first 48 h	- NRS 3, 24, 48 h - Non opioids cumulative consumption

OFA: Opiod free anestesia; OBA: opiod based anesthesia; TEA: thoracic epidural analgesia; NRS: numerical rating scale; h:hours; PACU: post anesthesia care unit; PONV: Postoperative nausea and vomiting; SBP: systolic blood pressure; INB: intercostal nerve block; SAPB: serratus anterior plane block; VATS: video-assisted thoracic surgery; PVB: paravertebral block; PCA: patient controlled analgesia; LOS: length of stay; PTI: pain threshold index; MAP: mean arterial pressure; HR: heart rate; ESPB:erector spinae plane block; RATS: robotic-assisted thoracic surgery.

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
