# Peer review of "Opioid Free Anesthesia in Thoracic Surgery: A Systematic Review and Meta Analysis"

_jcm, 2022, doi:10.3390/jcm11236955_

Round 1
Reviewer 1 Report
Dear authors,
Thank you for submitting a well conducted and presented systematic review with meta-analysis. Please find below some comments, which I believe they will improve the soundness of your work.
General comments:
• Please adjust scales of forest plots to more meaningful ranges, so that they can be viewed better by the readers.
• Please, do not use expressions like "we (they) found no difference" (there is always a difference found in a study between groups, the results are never identical). You can use instead "the differences were statistically non-significant".
Specific comments:
• Line 66: Please define LOS. It is the first time it appears here.
• Line 65: Could not find the registration number in PROSPERO website.
• Line 82: Please include your search strategy in the main document and not as an appendix.
• Line 109: Why have you calculated risk ratio and not odds ratio?
• Lines 116, 118, 120, 121, 157, 158, 162, 173, 185, 186: Please correct "I2" to "I2".
• Line 129: Please correct "five" to "six".
• Line 132: Is reference [21] correct? Should be [20]?
• Lines 133-136: The study by Bello et al. (2019) [15] was the only one with an open surgical technique, while the other ones were minimally invasive. That means that the analgesic requirements were significantly different between these two surgical techniques. This fact may have affected your results and should be appropriately addressed in the statistical analysis (forest plots, sensitivity analysis may be required) and in the discussion sections.
• Table 1: This table has been iterated. Please correct. Also, please correct the citation numbers of the included numbers, they are wrong. Please, correct the citation [20]; is it "Guangquan" or "An"?
• Line 159: PaO2, pH, SpO2 are not relevant to your study, please remove.
• Figure 5, 6, 7: High heterogeneity (95%-100%) may require a sensitivity analysis to discriminate the source of this heterogeneity.
• Figure 7: Maybe it should be omitted. It does not add much information, since it includes only two studies.
• Line 198: Please add "systematic review and" before "meta-analysis".
• Line 211: Please replace "Although" with "Despite".
• Line 218: Please replace "places" with "renders".
• Line 221: Please replace "were due to" with "were associated with".
• Lines 261-271: Please include in your limitations the very small number of included studies and the case of Bello et al. (2019) [15].
Reviewer 2 Report
This study is set up as a systemetic review and meta-analysis on OFA vs OBA in thoracic surgery. Is is an important and actual topic deserving much attention, regarding the undesired effects of opioids, also post-surgically. So, this makes this study very interesting. However this study has many serious flaws. First a meta-analysis is done on a small number of studies, whereas the number of included studies is confusing, is it five or is it six? Second, in the meta-analysis there is 1 RCT and 4 cohorts studies. Combining these types of studies in the same meta-analysis is to my opinion a major methodological problem, inducing a serious bias. Furhermore there are some inconsistencies. Therefore, despite the interesting topic, this study is to my opinion not publishable in its actual form.
Summary: clearly written
Introduction
40-41 Analgesia can be achieved either by using high doses of hypnotic agents
Is analgesia achievd by high doses hypnotics? This does not meet the idea of balanced anesthesia. Furhermore in modern anesthesia hypnotics are not use for obtaining analgesia. I suggest this should be changed.
53-55 Analgesia can be achieved either by using high doses of hypnotic agents
Please add a ref for his statement.
64 We performed a systematic review and meta-analysis
To my opinion you can not combine differents types of studies in a single meta-analysis, because of the quality differences of the type of studies leading to the risk of bias in processing the results in a meta-analysis. In this study 1 RCT and 5 observational cohort studies are combined in the meta-analysis. This is to my opinion a strong weakness in this study. This study should be limited to a SR.
Materials and methods
102-105 When scores were not presented in a 0-10 point scale format, they were converted (mean and SD) using the appropriate ratio. In particular, when reported on a scale 0-3, we consider 2 as the medium score (5), 0 as a minimum (0), and 3 as a maximum (10).
Although I understand reasoning for conversion of various types of pain scores to one, this also induces a risk of bias. Please comment on this.
Results
129-130 Five studies with a total of 802 patients met the inclusion criteria and were included (Figure 1).
In the introduction 904 patients are mentioned. Please comment on this discreipancy.
130-133 All studies were conducted between May 2016 and July 2019. Three studies were conducted in Europe [15, 16, 17], one in Australia [18], one in the USA [19], and one 131 in China [20] (Table 1). Only one study was randomized controlled [21], while the others were observational retrospective studies.
Six studies are described in the text as being included which in contrast with table 1. Please comment on this.
132-133 Only one study was randomized controlled [21], while the others were observational retrospective studies.
This is the study of Frauenknecht et al. Acoording to ref 21. However, this study is not included in te 5 or 6 (what is the correct number?) studies hat are included in tis study.
Figre 1. After 312 records being excluded after identifying 375 still 375 records are reviewed for title and abstract. Is this correct?
Table 1
- The table presents double the data; please delete one half of the table
- While it was mentioned that in the PRISMA analysis 1 RCT and 4 cohort studies were included in the PRISMA flowchart are finally 6 studies presented. This seems not to be correct. Please comment on that.
- I prefer the number of patients per study be presented in this table
- Number of excluded titles and abstracts is also not showed.
- Furthermore, adressing primary and secondary outcomes as defined in this SR/Meta-analysis are not presented either. This should to my opinion be extracted as well and presented in table 1.
- Bello et al as example: primary outcome is epidural ropivacaine consumption. What do we know about postoperative opioid consumption?
- Another problem is that OFA and OBA regimens are presented but post-operative analgesia regimens are not addressed.
Figure 2 Again combining a RCT with chohort studies and weighting ROB in one single attempt is methodologically incorrect, inducing a serious risk of bias.
Figure 3: In the Forest plots numbers of inckuded patients are named Total. I suggest to replace ‘Total’ by N, which is common use in presenting data referring to subjects.
I also suggest to present the p-values in the Forest plots.
LOS recovery room is not addressed anymore, while it is mentioned as secondary outcomes. Can authors comment on that?
Discussion
The 6th Chinese study mentioned in the first paragraph of the results, is not addressed at all. This is really confusing to me, as is also the number of finally included studies. Please comment on this.
Because of the methodology induced bias conclusion is too firm.
